# A Review and Follow-Up of Uterine Smooth Muscle Tumours of Uncertain Malignant Potential (STUMP): A Case Series and Literature Review

**DOI:** 10.3390/diseases11030099

**Published:** 2023-07-31

**Authors:** M. V. Lapresa-Alcalde, M. J. Ruiz-Navarro, M. Sancho de Salas, A. M. Cubo

**Affiliations:** 1Department of Obstetrics and Gynecology, Hospital Virgen de la Concha, 49022 Zamora, Spain; 2Department of Pathological Anatomy, Hospital Universitario de Salamanca, University of Salamanca, 37008 Salamanca, Spain; 3Department of Obstetrics and Gynecology, Hospital Universitario de Salamanca, University of Salamanca, Institute for Biomedical Research of Salamanca (IBSAL), 37007 Salamanca, Spain

**Keywords:** smooth muscle tumours, leiomyoma, leiomyosarcoma, atypical leiomyoma, uncertain malignant potential

## Abstract

Objectives: to analyse the clinical–pathological characteristics, treatment, and evolution of uterine smooth muscle tumours with uncertain malignant potential (STUMP) diagnosed in the Salamanca University Hospital with the implementation of the 2014 WHO criteria. Materials and methods: a retrospective descriptive study of patients diagnosed with STUMP from January 2015 to March 2023 at the Salamanca University Hospital. Demographic data, preoperative clinical data, treatment, complications, therapeutic results, anatomopathological findings and recurrence time were obtained. Results: a total of four patients were identified and included in the study. The mean age at diagnosis was 48 years (range 36–67). The surgical indications were abnormal uterine bleeding, compressive symptoms, and the growth of a pelvic mass suspected to be a degenerated myoma from the residual cervix after a subtotal hysterectomy 6 years earlier. In all cases, a laparotomic procedure was performed. A total hysterectomy, sub-total hysterectomy, and the excision of the cervix with STUMP localization were accomplished in two, one, and one patient, respectively. The mean diameter of the tumour pieces was 13 cm (range 8–17 cm), with a mean volume of 816 cc (range 234–1467 cc). The mean follow-up was 47 months, with no recurrence to date. Conclusions: STUMPs are a heterogeneous group of tumours with a difficult-to-predict clinical evolution. In most cases, their diagnosis is histological after performing surgery for suspected leiomyoma. Due to their low incidence, there are no specific guidelines for their treatment and control. However, considering their potential risk of recurrence and metastasis, it is advisable to maintain six-monthly controls for 5 years and then annual controls for 5 years more.

## 1. Introduction

Uterine smooth muscle tumours have, historically, been classified into benign leiomyomas and malignant leiomyosarcomas according to the degree of cytological atypia, mitotic activity, and other molecular tissue markers [1]. However, there is a spectrum of borderline tumours, including variants of mitotically active, cellular, and atypical leiomyomas, as well as Smooth Muscle Tumours of Uncertain Malignant Potential (STUMP) [2]. This term was introduced by Kempson in 1973 [3] and, according to the WHO, should be used for fibroids that cannot be unequivocally and histologically diagnosed as being benign or malignant [4]. However, patients who are affected by uterine STUMP present symptoms similar to those in leiomyoma and leiomyosarcoma, such as abnormal uterine bleeding, anaemia, chronic pelvic pain, pelvic mass, menorrhagia, or infertility. Nevertheless, some patients can be asymptomatic, leading to a delayed diagnosis that might lead to a worse prognosis [5,6].

It is thought that STUMP may be the transition from leiomyoma to leiomyosarcoma or a sometimes underestimated low-grade leiomyosarcoma [7,8]. Most reports of uterine STUMP have found similar age of diagnosis to that of fibroids or sarcomas, and the median age of this is between 40 to 50 years [5]. Compared to sarcomas, they have a better prognosis, but their biological potential is uncertain, as recurrence, malignization, and metastases are possible until many years later (11% mean recurrence rate diagnosed after a mean time of 51 months from initial diagnosis) [8]. It has been suggested that recurrence is more likely at younger ages [6].

A careful histopathological study is needed to confirm the right diagnosis of STUMP, due to the unreliability of preoperative imaging techniques in differentiating between leiomyoma, STUMP and leiomyosarcoma [5]. There are no clinical guidelines on the management of STUMP, so the clinical approach to the diagnosis, treatment and control of its recurrence is based on observational data [7].

This study aims to analyse the clinical–pathological characteristics, treatment, and follow-up of STUMPs diagnosed in the Salamanca University Hospital from 2015 to 2023.

## 2. Materials and Methods

A retrospective descriptive study of patients diagnosed with STUMP from January 2015 to March 2023 at the Salamanca University Hospital was performed. Demographic data, preoperative symptoms, treatment, complications, therapeutic results, anatomopathological findings, and recurrence time were obtained from the patients’ records.

The tumour volume was calculated according to the size of the surgical pieces, applying the volume of an ellipse formula (length × width × depth × 0.5233).

Recurrence was defined as a diagnosis of STUMP or leiomyosarcoma at least 6 months after surgery. A diagnosis of leiomyoma was not considered a recurrence. This study was approved by the Salamanca University Hospital Ethics Commission.

## 3. Results

In total, 4 patients with a pathological diagnosis of STUMP out of 915 women undergoing surgery for uterine leiomyomas during the study period were included.

Their characteristics are presented in Table 1. The mean age was 48 years (range 36–67 years). Only one patient (case no 4) was menopausal. The reason for surgery was abnormal uterine bleeding in the first patient. Patient 2 showed compressive symptoms, fever, a rise in inflammatory parameters, and suspicion of a haemorrhagic degeneration of the uterine myoma by Computed Tomography. The third case also presented compressive symptoms, together with an increase in the size of the fibroid, despite treatment with Ulipristal Acetate in the previous 4 months. The surgical indication in the fourth case was the growth of a pelvic tumour suspected to be to be a myoma of residual uterine cervix after a presumed total hysterectomy performed 6 years earlier. A total hysterectomy was performed on two patients (cases no 2 and 3) and a subtotal hysterectomy was initially performed on patient no 1, although after the diagnosis of STUMP, a vaginal cervicectomy was carried out. In case 4, a resection of the tumour and the remaining cervix was performed. The average diameter of the tumour pieces was 13 cm, with an average volume of 816 cc.

Two patients had severe surgical complications. Case 3 suffered a distal injury of the left ureter, requiring ureteral neo-cystostomy 7 months later. Case 4 was complicated by a perforation of the sigmoid colon, requiring colon resection and termino-terminal (end to end) anastomosis. During the first surgery day, a left ureteral lesion was diagnosed and followed by ureteral neo-cystostomy, according to the Lich-Gregoir technique. A double J catheter was left and retired 2 months later.

The mean follow-up was 47 months, without diagnosis of any recurrence. The patients had a check-up one month after surgery and every six months for up to five years.

## 4. Discussion

Mesenchymal smooth muscle cell tumours are the most common type of uterine neoplasia. They include leiomyomas and their subtypes, mesenchymal smooth muscle tumours of uncertain malignant potential (STUMP), and leiomyosarcomas [9]. The main problem in most cases is that a diagnosis of certainty is obtained via an anatomopathological study, and this is not possible until the patient undergoes surgery. For this reason, it is of interest to continue investigating the diagnosis prior to surgery to identify masses with suspected malignancy [6].

At present, the imaging studies available to us do not show a clear difference between STUMPs or leiomyomas and leiomyosarcomas. Ultrasonography is the diagnostic tool most commonly used by gynaecologists. However, in the studies found in the literature, ultrasound is not effective for the differentiation of these tumours [5]. A recent study by Russo et al. showed that the age of the patient combined with the size and the intralesional and circumferential vascularity of the tumour could help to differentiate tumours [10]. Magnetic resonance imaging is another tool, which, theoretically, has a superior soft tissue resolution to ultrasonography. Some studies have indicated that contrast-enhanced magnetic resonance imaging could offer a better preoperative differentiation between STUMPs or leiomyomas and leiomyosarcomas [5]. Positron emission tomography with 18-FDG in a 2018 study showed a “hollow ball” sign on the FDG PET, which corresponded to areas of coagulative necrosis of the tumour in STUMPs and leiomyosarcomas, but this is not found in leiomyomas [11].

According to the 2014 WHO criteria [9], the pathological diagnosis of STUMP is based on the presence of coagulative necrosis (cases 1 and 4), the finding of 24 mitoses per field and moderate focal atypia (case 2), and diffuse moderate atypia (case 3) (Figure 1 and Figure 2).

The Stanford criteria for the diagnosis of leiomyosarcoma reported by Bell et al. include at least two of the following parameters: moderate–severe diffuse cytological atypia, tumour necrosis, and at least 10 mitoses per high-magnification field (≥10 MF/10 HPF) [12]. If a tumour exhibits any combination of these three characteristics, but does not meet the Stanford criteria for the diagnosis of leiomyosarcoma, it can be diagnosed as STUMP [1]. The diagnosis of benignity (leiomyoma) is characterized by an absence of atypia and tumour necrosis and ≤4 MF/10 HPF. Some variants of leiomyoma include more than 5 and less than 19 MF/HPF (mitotically active leiomyoma), or cytological atypia without tumour necrosis and <10 MF/10 HPF (atypical leiomyoma) [8], although this term is not universally accepted by pathologists. At present, “bizarre leiomyomas” or “leiomyomas with bizarre nuclei”, are classified as benign myomas [6].

Currently, the diagnosis of STUMP is based on criteria approved by the WHO in 2014 [9] (Table 2):Tumours with focal or moderate-to-severe multifocal atypia, without cell tumour necrosis and a mitotic count less than or equal to 10 mitotic figures per field.Tumours with moderate-to-severe diffuse atypia and a mitotic count of less than 10 mitotic figures per field.Tumours with cell necrosis, mild or absent atypia, and a mitotic count less than 10 mitotic figures per field.Tumours without cell necrosis, mild or absent atypia, and a mitotic count greater than or equal to 15 mitotic figures per field.

Among the three major criteria for establishing the biological potential of uterine tumours (cytological atypia, mitotic index, and coagulative tumour necrosis), the one most strongly associated with malicious behaviour is the latter [6]. In the absence of cell necrosis, the factor that determines the tumour behaviour is the mitotic index. Other factors have been related to tumour recurrence: after primary surgery by morcellation, it could be possible to disseminate the tumour if an unprotected morcellation is used [5]. Other less important prognostic factors include the infiltration of surgical borders, postmenopausal patients, and a tumour size greater than 3 cm [8].

The expression of immunohistochemical markers could be helpful in differentiating these tumours. Ki 67, p53, and p16 have been studied either individually or in combination. Steroid hormone receptors, such as the estrogen receptor and progesterone receptor, could be present in STUMP and leiomyomas. The results of studies have concluded that the expression of immunohistochemical markers in STUMP is similar to that in leiomyomas. However, the immunohistochemical markers in leiomyosarcoma are different from the others. This fact allows a research line for the study of new specific markers that can improve the accuracy in differentiating STUMP from other smooth muscle tumours [5].

The incidence of STUMP is difficult to estimate. Among women undergoing a hysterectomy or myomectomy for a presumed diagnosis of leiomyoma, 0.01% receive a diagnosis of STUMP. An average age at diagnosis of 43 years has been suggested (similar to our study) [7].

In our case series, as in the published literature [1,6,8], suspicions of STUMP had not been established prior to the interventions, since the signs and symptoms were similar to those of leiomyomas (abnormal uterine bleeding, pelvic pain, and pelvic compression). Only in one case was there suspicion of malignancy due to rapid growth and the appearance of degeneration in radiology (case no. 4).

Given the rarity of this tumour and the scarcity of the published series, there are no clinical guidelines for the management of STUMP, so the clinical approach in terms of the diagnosis, treatment, and control of its recurrence is based on observational data [7]. For patients who have completed their fertility desire, a total hysterectomy with or without bilateral salpingo-oophorectomy represents a conventional surgical treatment for avoiding possible recurrence [5,6,13]. However, in women who wish to preserve their fertility, a myomectomy may be considered, taking into account its slow growth. Although published data on gestations in women with these tumours are scarce [1,2], some studies have shown promising results, with success rates of 70–80% [1,14]. No adjuvant to prevent recurrences was found in the literature [6]. Other therapeutic approaches, such as uterine artery embolization, do not seem to be recommended [15].

The high rate of severe surgical complications in our case series could have been due to the large size of the pelvic tumours. The cases that suffered surgical complications (no 3 and 4) had a larger volume than a 26-week pregnant uterus.

The estimated recurrence rate of this kind of tumour is between 3.7 and 27%, with an average time of 51 months [2,7,13]. This variability can be attributed to the scarce number of studies and cases included in each study, as well as the disparity of the criteria for defining recurrence [1,2,6]. In the series published by Sahin et al. (57 cases), the subserosal location of the STUMP was significantly associated with a higher number of recurrences [14]. Some authors have suggested that immunohistochemical characteristics (positivity for p16, p53, MIB-1, bcl-2, estrogen, and progesterone receptors) or serological markers (CA 125 and HE-4) could be useful for identifying worse prognoses and a higher risk of recurring tumours [6,13,16]; however, the predictive value of these elements is not well established and has not been studied in a large series so far [1]. Croce et al., in a European multicentre study, proposed a genomic analysis using array-CGH to identify the most aggressive STUMPs. Although these tumours had a higher genomic index, the differences were not significant in terms of survival [17]. When recurrence occurs, the best treatment is a surgical resection if possible. A few studies have shown individual cases with adjuvant treatment. As for hormone therapy, the agents most commonly mentioned are progesterone, aromatase inhibitors, and gonadotropin-releasing hormone analogues. Some studies have reported cases of recurrences and metastases in the lung, lymph nodes, or abdominal cavity, which have been successfully treated with monotherapy or combined therapy with aromatase inhibitors, gonadotropin-releasing hormone analogues, or progesterone and estrogen receptor antagonists. However, this could be related to the fact that these patients had progesterone-receptor-positive and estrogen-receptor-positive STUMPs. Therefore, in patients who do not have these receptors, hormone therapy may not be effective. The chemotherapy agents used in these reports included doxorubicin, cisplatin, gemcitabine, docetaxel, and ifosfamide. These chemoagents could be used alone or in combined therapy, although there is no consensus on this point either. No studies have been found on radiotherapy for the recurrence of STUMP [5].

In our series, there were no recurrences, although the short follow-up period (mean of 46.5 months with a range of 7–64 months) have interfered with this result.

Given the difficulty of establishing reliable recurrence rates and their potential for metastasis, it is important to maintain patient adherence and adequate controls. Although there is no consensus regarding the necessary controls after surgery, the most accepted standard follow-up consists of six-monthly clinical reviews for 5 years, followed by annual reviews for 5 more years and an annual MRI [6,13]. Furthermore, these controls should include gynaecologic examinations, abdominopelvic ultrasonography, and chest radiography. For patients who decide to delay surgery to preserve their fertility, an annual MRI can be performed, replacing the CT scan [5]. Since STUMPs are rare tumours, it has been proposed to refer cases to specialized centres for proper diagnosis and follow-up [18]. This would make it possible to optimize care, reduce mortality, and improve the management of this type of rare tumour [2,18].

## 5. Conclusions

STUMPs are a rare and heterogeneous group of tumours with difficult-to-predict clinical behaviour. The pathologic criteria for STUMP diagnoses are not defined and it is necessary to create a consensus to resolve this problem. In most cases, its histological diagnosis is achieved after surgery for a suspected leiomyoma. The challenge is to find diagnostic tests that help us to suspect these tumours before surgery. Although immunohistochemical studies may help in differentiating these tumours, more research is required to improve their diagnosis. A total hysterectomy is the standard and most accepted surgical treatment, although a myomectomy may be considered for women who wish to preserve their fertility. To date, neither immunohistochemical characteristics nor serological markers seem to be useful in identifying worse prognoses and a higher risk of recurring tumours. Due to STUMP’s potential for recurrence and metastasis, long-term follow-ups seem necessary.

## Figures and Tables

**Figure 1 diseases-11-00099-f001:**
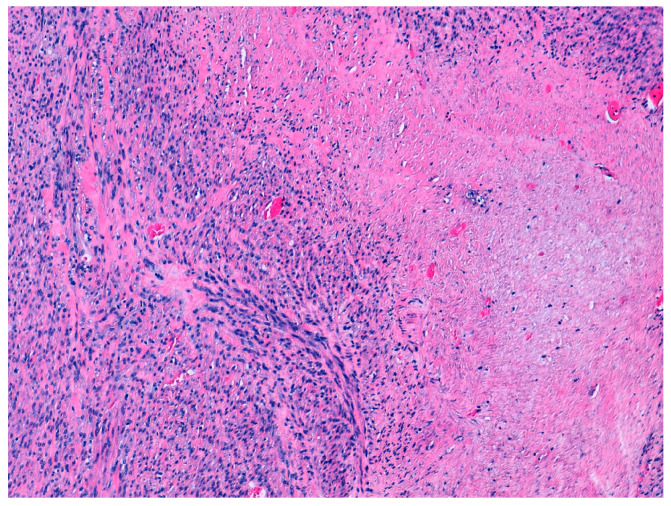
Hematoxilin and eosin staining (10×). Focal cell necrosis. Case 1.

**Figure 2 diseases-11-00099-f002:**
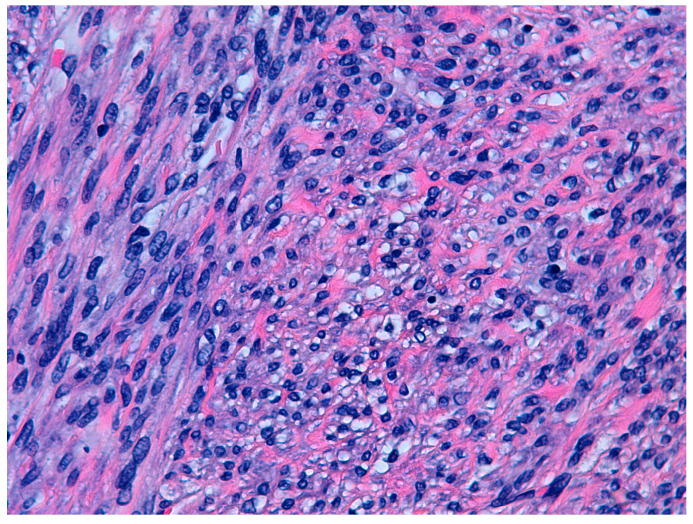
Hematoxilin and eosin staining (40×). Cytological atypia. Case 2.

**Table 1 diseases-11-00099-t001:** Clinical–pathological characteristics of the patients included in the study.

	Case 1	Case 2	Case 3	Case 4
Year	2016	2017	2018	2018
Age	36	38	50	67
G/P	2/2	1/0	2/2 (caesarean sections)	3/3
Menopause	no	no	no	yes
Symptoms	Abnormal uterine bleeding and dysmenorrhoea	Abdominal pain, constipation, and fever	Abdominal pain, urinary incontinence, tumour growing, and renoureteral colic pain	Abdominal pain and fast-growing pelvic mass
Surgery	Subtotal hysterectomy + bilateral salpingectomy	Total hysterectomy + bilateral salpingectomy	Subtotal hysterectomy + bilateral adnexectomy	Tumour and remaining cervix resection
Previous treatment	Ulipristal acetate	-	Myomectomy, 1998. Ulipristal acetate	Hysterectomy + bilateral adnexectomy, 2002
Tumour size (cm)	10	8	17	17
Volume (cc^3^)	293	234	1272	1467
Atypia	Mild and focal	Moderate/intense and focal	Moderate and diffuse	-
Tumour necrosis	Positive	-	-	Focal
Mitotic figures/10 HPF	4	24	5	5–10
p16 expression	Focal	Focal	Focal	Positive
p53 expression	-	-	-	Focal
Post-surgery treatment	Remaining cervix vaginal resection		Percutaneous nephrostomy.	
Follow-up (months)	64	60	55	7
Recurrences	No	No	No	No

G/P: pregnancies/type of birth.

**Table 2 diseases-11-00099-t002:** Pathological characteristics of STUMPs and their published recurrence rates. Adapted from “WHO Classification of Tumours of Female Reproductive Organs” [9].

Tumour Cell Necrosis	Moderate-to-Severe Atypia	Mitotic Count (per 10 HPF *)	Mean Mitotic Count in Tumours with Recurrence (Per 10 HPF *)	Cases with Recurrence
Absent	Focal/multifocal	<10	4 (range 3–5)	13.6%
	Diffuse	<10	4′3 (range 2–9)	10.4%
Present	None	<10	2′8 (range 1–4)	26.7%
Absent	None	≥15	Not applicable	0%

* HPF: high power field.

## Data Availability

Data is unavailable due to ethical restrictions.

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
