# Peer review of "A Review and Follow-Up of Uterine Smooth Muscle Tumours of Uncertain Malignant Potential (STUMP): A Case Series and Literature Review"

_diseases, 2023, doi:10.3390/diseases11030099_

Round 1
Reviewer 1 Report
General Comments
This is an interesting paper addressed to establish the clinical outcome of 4 patients suffering from a rare disease affecting myometrium. However, the manuscript requires major revisions. The English style and grammar should be reviewed by a native English speaker with experience in medical papers. A Results section should be constructed and integrated between methods and discussion sections. The overall number of interventions accomplished to treat smooth muscle uterine tumours during the 8 years study-period must be reported in order to define the prevalence of STUMP in that Hospital experience. In order to improve the contents and the reading of the manuscript, the production of histology photograms, if available, is warranted.
Title: The title is misleading. This is an observed case-series. I believe it should be changed. “Uterine Smooth Muscle Tumours of Uncertain Malignant Potential: Case Series and Literature Review”, at the judgement of Authors, seems more appropriate.
Abstract Section
Line 18-20: “(range 36-67)”, “(case n°1)”, “(cases 2 and 3)”, “(case n°4)” are unnecessary.
Line 20-21: The sentence “In cases 1 and 3 …clinical improvement” is of no relevance in abstract section.
Line 21-22: The sentence should be modified to: “In all cases a laparotomic procedure was performed. Total hysterectomy, sub-total hysterectomy and excision of the cervix with STUMP localization were accomplished in 2, 1 and 1 patient, respectively.”
Introduction Section
Line 36-37: The sentence must be modified: “… mitotic activity, presence of necrosis and other molecular aspects (1)”. Other molecular aspect should be better defined such as “molecular tissue markers” or “biological markers”.
Line 38: The term “intermediate” appears redundant and it should be deleted
Line 40: The term “described” is not appropriate. It should be changed to “firstly used” or “introduced”
Line 42: The sentence should be changed to “Patients who are affected by…present symptoms similar to…
Line 44: I suggest: “Nevertheless, some patients can be asymptomatic, leading to a delayed diagnosis that might worsen the prognosis”
Line 47-48: I suggest: “… may be the transition from leiomyoma to leiomyosarcoma or a sometime underestimated low-grade leiomyosarcoma.” Moreover, “…and they have a slow growth” should be deleted.
Line 52: I suggest; … (11% mean recurrence-rate diagnosed after a mean time of 51 months from initial diagnosis).
Line 55-57: These two sentences should be changed. I suggest: “A careful histopathological study is need to confirm the right diagnosis of STUMP, due to the unreliability of preoperative imaging techniques in differentiating between leiomyoma, STUMP and leiomyosarcoma (5)”. Moreover, what the author means by the bracketed “Taiwan” term?
Line 62: The term “evolution” should be changed with “follow-up”
Material and Methods Section
Line72-73: The sentence “Loss of follow-up…” is not understandable.
The text between lines 75 to 92 and lines 97 to 104 should be included in a Results paragraph.
Line 75: Table 1 must be completely reviewed to provide a more synthetic and understandable editorial format. What the author means by the term “Menstrual formula”?
Line 75: “Four patients with a pathological diagnosis of STUMP out of …. women undergoing surgery for uterine leiomyomas during the study period were included”.
Line 76: “Only one patient (case n°4) was menopausal”
Line 77-78. The sentence “Concerning… two gestations” should be deleted. The data are derived from Table 1. “The indication for surgery was…”
Line 79: “Patient 2 showed compressive symptoms, fever, rise of inflammatory parameters and suspicion of hemorrhagic degeneration of uterine myoma by Computed Tomography”.
Line 83: (abdominal pain…retention) should be deleted
Line 85-86: “The surgical indication…”. “to be a myoma of residual uterine cervix after a total hysterectomy performed 6 years earlier.
Line 87-88: “Total hysterectomy was performed in two patients… patient n° 1, though after the diagnosis of STUMP a vaginal cervicectomy was carried out. In the case 4…”
Line 94: “…, the pathological diagnosis of STUMP was based on the presence of…. the finding of 24 mitoses…”
Line 97: “Two patients…”
Line 98-101: “… ureteral neo-cystostomy 7 months later.” “The case 4 was complicated by a perforation of the sigmoid colon, requiring colon resection and termino-terminal anastomosis.” “During the first operative day a left ureteral lesion was diagnosed and followed by ureteral neo cystostomy according to the…”
Line 103-104: The timing and medical criteria used in follow-up must be stated. What the author means by (range 64-7)? What the author means by “All following revision were normal”?
Discussion Section
Although the presentation is in keeping with the clinical issues proposed by a diagnosis, management and follow-up of STUMP, this section is very difficult to read in the present format. It requires a supervision of a physician experienced in scientific literature production, a slender presentation and an extensive review of style and English grammar. I individuate the need of substantial review of discussion section in lines 108-109, 150-153, 158-159, 166-168, 174-177, 184-185, 186-187, 195-196, 200-201, 206-207. The exceedingly high rate of severe surgical complications requires a comment.
The English language and style of manuscript must be reviewed by a native english speaker experienced in medical papers and by a physician with expertise in scientific literature production.
Reviewer 2 Report
This article describes the cases of uterine STUMP with follow up data and review. It summarizes uterine STUMP and provide useful information. However, there are several points that need to be revised.
1. Line 56, 122, and 180: Why is the word (Taiwan) placed in this sentence? Please explain the meaning.
2. Line 119: “Magnetic resonance imagin” should be “Magnetic resonance imaging”.
3. Line 201: “aromatasa” should be “aromatase”.
4. There are several spelling and grammatical errors in the article.
5. Histological picture of the cases should be presented.
There are several spelling and grammatical errors in the article. So, please edit the English language.
Round 2
Reviewer 1 Report
Dear Author
1.The English grammar and style must be further improved, particulary in the Discussion section.
2. Throghout the manuscript, I suggest to reduce the redundant similar sentences about the low prevalence of STUMP and the lack of clinical data.
3. Table 1. The case 4 was diagnosed as STUMP arising from the residual cervix in a patient previously treated by total hysterectomy. This mistake needs to be clarified/modified as well as in the Results section, line 82 (...total hysterectomy?).
4. Results section, lines 88-89. This still incomplete sentence must be placed at the beginning of the Results section. Possibly, it should include all variant of leiomyomas and the leiomyosarcomas treated during the eight years study period.
I believe the English grammar and style must be further improved.
Author Response
-English language has been checked by a native colleague.
-Redundant similar sentences about the low prevalence of STUMP and the lack of clinical data have been removed.
-Case 4 underwent an presumed total hysterectomy (line 85). There was probably some residual tissue left unresected that 6 years later led to a STUMP.
-Total number of the patients undergoing surgery for uterine leiomyomas has been inserted.
Reviewer 2 Report
In this version of the article, quality was improved but there are still uncorrected spelling and grammatical errors. Authors are advised to check every word carefully and to revise it again.
1. Line 88: “Four patients with a pathological diagnosis of STUMP out of …. women undergoing 88 surgery for uterine leiomyomas during the study period were included” Please insert total number of the patients in this sentence.
2. Representative histological pictures should be presented.
Please correct spelling and grammatical errors.
Author Response
-Total number of the patients undergoing surgery for uterine leiomyomas has been inserted.
-Representative histological pictures have been added.
-English language has been checked by a native colleague.